# Sex and Diet-Related Disparities in Low Handgrip Strength among Young and Middle-Aged Koreans: Findings Based on the Korea National Health and Nutrition Examination Survey (KNHANES) from 2014 to 2017

**DOI:** 10.3390/nu14183816

**Published:** 2022-09-15

**Authors:** Inhye Kim, Kumhee Son, Su Jin Jeong, Hyunjung Lim

**Affiliations:** 1Department of Medical Nutrition, Graduate School of East-West Medical Science, Kyung Hee University, Yongin 17104, Korea; 2Research Institute of Medical Nutrition, Kyung Hee University, Seoul 02447, Korea; 3Statistics Support Part, Medical Science Research Institute, Kyung Hee University Medical Center, Seoul 02447, Korea

**Keywords:** muscle strength, diet, Korea

## Abstract

“Possible sarcopenia” may be defined as a low muscle strength assessed by handgrip strength (HGS) by sex. We examined the sex-specific association between low handgrip strength (LGS) and dietary factors for the prevention of sarcopenia in young and middle-aged Koreans. We used data from the 2014–2017 Korea National Health and Nutrition Examination Survey of 11,635 Korean adults with LGS and normal handgrip strength (NGS). The relationship between dietary factors, e.g., nutrients, foods, and dietary patterns, and HGS was evaluated by multivariate logistic regression analyses. In men, the LGS group had a higher proportion of energy from carbohydrates and a lower proportion of energy from proteins than the NGS group. The LGS group had lower protein, niacin, phosphorus, and iron densities in their diet than the NGS group. The odds of having LGS increased as intake of vitamin B1 (odds ratio (OR) 2.916, 95% confidence interval (CI) 1.265–6.719), niacin (OR 2.286, 95% CI 1.095–4.774), phosphorus (OR 2.731, 95% CI 1.036–7.199), and iron (OR 2.591, 95% CI 1.102–6.088) decreased. In women with LGS, the odds of insufficient protein intake (OR 1.976, 95% CI 1.248–3.127) was significantly higher. This study suggests that adequate intake of protein, vitamin B1, niacin, phosphorus, and iron is beneficial for maintaining HGS.

## 1. Introduction

As South Korea is one of the fastest aging countries and is predicted to enter “Super-aged society” by 2026, people have much interest in prevention and management of age-related health problems [1]. A notable phenomenon associated with aging is a change in body composition. A decrease in muscle mass and an increase in body fat begin to appear in middle age, consequently leading to a loss of muscle strength and power in old age [2,3]. Age-related loss of muscle strength is estimated to be approximately 1.5% annually [4], and muscle weakness is considered an underlying factor contributing to ill health [5,6]. According to the report on sarcopenia prevalence of Korea, the prevalence of sarcopenia is 19.2%, 29.1%, and 42.3% among the 20 to 39, 40 to 64, and over 64 years age groups, respectively. Therefore, muscle health has become a problem that cannot be ignored not only in the elderly but also in the young and middle-aged population [7,8].

Handgrip strength (HGS) has recently attracted attention as a new subject in young and middle-aged population in Korea because HGS is associated not only with sarcopenia, but also with chronic diseases such as cardiovascular disease, diabetes, and metabolic syndrome [6,9]. In recent studies, HGS was found to have negative association with the prevalence of metabolic syndrome [5,10]. Skeletal muscle plays an important role in energy metabolism and is involved in whole-body metabolic homeostasis by secreting myokine, which is related to metabolic syndrome management [11,12]. Severe obesity in young and middle-aged adults was associated with sarcopenia and poor muscle quality [13]. Considering the fact that the incidence of obesity and metabolic syndrome in young and middle-aged population is increasing [14], it is important to maintain adequate HGS in young and middle-aged population.

The cutoff values of the diagnostic criteria for low handgrip strength (LGS) vary by country and sex. The Asian Working Group for Sarcopenia (AWGS) 2019 defined low muscle strength as HGS <28 kg for men and <18 kg for women, regardless of age. In addition, they introduced “possible sarcopenia”, defined by either low muscle strength or low physical performance, specifically for use in primary health care or community-based health promotion, to enable earlier lifestyle interventions [15]. It is thought that the prevention of muscle weakness is possible through lifestyle factors [16].

HGS can be influenced by several factors, including age, sex, height, weight, nutritional status, education level, and physical activity [9]. In particular, nutrition has been shown to be a key component in LGS [6]. A previous study showed that total energy, protein and polyunsaturated fatty acid were positively correlated with HGS [9]. Micronutrients including B vitamins, vitamin D, and E and Fe intake were also associated with HGS [17]. Recently, it has been difficult to determine the correlation between single nutrients and muscle strength; several studies using dietary pattern analysis have been conducted [18,19]. Especially, the Mediterranean dietary pattern has been associated with HGS [19]. However, the Korean diet is characterized by higher carbohydrates and lower fat than the Western diet; thus, it is hard to compare the Korean diet with the Mediterranean diet.

Despite age- and sex-specific differences in HGS, there is a dearth of information on the association of dietary factors and LGS in Korean adults according to sex. Moreover, as previous studies focused mainly on the elderly, a study of HGS including the young and middle-aged population is needed. The purpose of this study was to identify exploratively the sex-specific relationship between dietary factors and HGS according to sex among young and middle-aged Koreans, using data from the Korea National Health and Nutrition Examination Survey (KNHANES) from 2014 to 2017.

## 2. Materials and Methods

### 2.1. Data Source and Study Population

The data used in this study were collected from the 2014–2017 KNHANES. KNHANES is a series of nationwide, population-based, cross-sectional survey by the Korea Centers for Disease Control and Prevention in Cheongju, South Korea, since 1998 [20]. This survey is based on data from Statistics Korea’s Population and Housing Census and Public Price of Apartment Houses in Daejeon, South Korea. Two hundred and eighty-two households in 192 regions are selected as probability samples for Koreans over the age of 1, and representative groups are selected as sample designs stratified by region, age, and gender. To represent all parts of the nation, two-stage stratified cluster sampling and rolling sampling survey is used. KNHANES consists of health examination, health interview, and nutrition survey. Well-trained medical staffs and interviewers perform this survey. Health examination consists of anthropometric measurements, body composition exam, blood chemistry, urinalysis, oral exam and eyesight test. HGS is measured for those aged 10 or older. Health interviews are self-written and consist of physical activity, sleep health, obesity, weight control, drinking and quality of life. The nutrition survey consists of the eating habits, food intake and food stability survey.

Information on 15,830 young and middle-aged Koreans aged 19–64 years who participated in the survey between 2014 and 2017 was collected. Thereafter, we excluded participants with missing data on HGS (*n* = 1880), extreme energy intake level (i.e., <500 or >5000 kcal/day) (*n* = 287), diseases can affect HGS such as diabetes mellitus (*n* = 639) [21], thyroid disorder (*n* = 264) [22], cancer (*n* = 189) [23], and musculoskeletal disorder (*n* = 842), and those who were pregnant (*n* = 94) [24]. In total, 11635 participants were included in our study. This study was conducted in accordance with the guidelines of the Declaration of Helsinki, and its protocol was approved by the Institutional Review Board of Kyung Hee University in Seoul, South Korea (KHSIRB-20-053 (EA)).

### 2.2. Assessment of Sociodemographic and Anthropometric Characteristics

Sociodemographic data of the participants included age, household income, residential area, alcohol consumption, smoking status, and physical activity. Household income was categorized into quartiles, and the residential area was classified as urban or rural. Alcohol consumption was categorized into no consumption (none), low consumption, and high consumption. High consumption was defined as those who consumed more than seven (men) or five (women) drinks per day, more than twice per week. Low consumption was defined as those consuming less than high consumption definition. Smoking status was categorized as nonsmoker (never smoked cigarettes), ex-smoker (smoked at least 100 cigarettes but do not currently smoke), or current smoker (smoked at least 100 cigarettes and presently smokes). Physical activity was categorized, according to the total metabolic equivalent (MET) score, as low (<600 MET-min/week), moderate (600–3000 MET-min/week), or high (>3000 MET-min/week) according to IPAQ guideline [25]. High means (i) vigorous-intensity activity on at least 3 days and accumulating at least 1500 MET-minutes/week or (ii) 7 or more days of any combination of walking, moderate-intensity or vigorous intensity activities achieving a minimum of at least 3000 MET-minutes/week. Moderate means (i) 3 or more days of vigorous activity of at least 20 min per day or (ii) 5 or more days of moderate-intensity activity or walking of at least 30 min per day or (iii) 5 or more days of any combination of walking, moderate-intensity or vigorous intensity activities achieving a minimum of at least 600 MET-min/week. Low means who do not meet criteria for moderate or high.

Height (Seca 225; GmbH & Co. KG, Hamburg, Germany), weight (GL-6000-20; Caskorea, Seoul, Korea), and waist circumference (Seca 201; GmbH & Co. KG, Hamburg, Germany) were measured by trained medical staff using standardized techniques and calibrated equipment. Body mass index (BMI) was calculated as weight (kg) divided by height squared (m^2^).

### 2.3. Measurements of Handgrip Strength

HGS was measured using a digital hand dynamometer (Digital Grip Strength Dynamometer, TKK 5401; Takei Scientific Instruments Co., Ltd., Tokyo, Japan). All participants were instructed to hold the dynamometer in an upright position with their arms at their sides and to squeeze the grip with full force for 3 s each time for each hand. An interval of 60 s was allowed between each strength measurement. According to the recommendations of the international working group on sarcopenia [26], the highest value of six attempts was used for data analysis.

The AWGS 2019 defined low muscle strength as HGS <28 kg for men and <18 kg for women. This study applied the same definition to categorize participants into LGS (<28 kg for men and <18 kg for women) group and normal HGS (NGS; ≥28 kg for men and ≥18 kg for women) group [15]. To increase the validity of the measured HGS, additional indicators related to HGS were used. Hand-grip-strength-to-weight ratio (HGSWR) was calculated as HGS divided by weight (kg) and multiplied by 100 [27]. Absolute HGS was calculated as the sum of the maximal HGS readings for each hand. Relative HGS was calculated as the absolute HGS divided by BMI [28].

### 2.4. Nutrients Intake Analysis

Dietary intake was assessed using the 24 h recall method a week after the health interview and examination including meals, snacks, and dietary supplements by trained interviewers. Nutrient intake was calculated using the Korean Foods and Nutrients Database of the Rural Development Administration. The adequacy of protein; vitamin B1, B2, and C; niacin; calcium; phosphorus; and iron intake was estimated using the estimated average requirement (EAR) in the Dietary Reference Intakes for Koreans 2015 (KDRIs) [29]. Inadequate protein intake indicated the proportion of individuals who consumed <0.73 g/kg/day, which is the EAR.

### 2.5. Dietary Pattern Analysis

The food groups were divided into 25 groups for the dietary pattern analysis. Common food groups were divided into 18 categories (grains, potatoes, sugars, beans, nuts, vegetables, mushrooms, fruits, meats, eggs, fish, seaweeds, dairy products, oils, beverages, seasonings, proceed foods, and other) per the Korean Nutrient Database. Because of the high intake of grains and their derivatives in Korea, we further divided grains into four subgroups based on their nutritional profiles: white rice, other whole grains, wheat flour and bread, and noodles. Dietary patterns were derived by factor analysis using the principal component analysis of the nutritional intake in grams per day.

### 2.6. Statistical Analysis

All analyses were performed separately according to sex. Continuous variables were reported as mean ± standard error according to PROC SURVEYMEANS procedure. Categorical variables were described as frequencies (*n*) and percentages (%) according to the PROC SURVEYFREQ procedure. Multivariate generalized linear regression according to the PROC SURVEYREG was conducted using the nutrient and food intake data. Dietary patterns were classified by tertiles according to the factor score. The number of factor patterns was calculated using an orthogonal varimax rotation method, with eigenvalues >1.3, and analysis of the scree plot. We performed Kaiser–Meyer–Olkin and Bartlett’s sphericity assumption. For the appropriate interpretation of dietary patterns, food groups with a factor loading of ≥0.25 (men) and ≥0.30 (women) were defined as significant contributors to dietary patterns. The analyses of the variance and chi-squared test (*p* < 0.05) were used to investigate characteristics according to sex. The lowest tertile of the food intake and dietary pattern was used as the reference for multiple logistic regression analysis.

Logistic regression analysis was used to investigate the association between the adequacy of dietary intake, food intake, dietary pattern, and LGS at 95% confidence interval (CIs) using the PROC SURVEYLOGISTIC procedure. All statistical analyses were performed using the PROC SURVEY in SAS version 9.4 (SAS Institute, Cary, NC, USA). All statistical analyses were two-tailed and considered significant at *p* < 0.05.

## 3. Results

### 3.1. Sociodemographic and Anthropometric Characteristics of the Participants

Table 1 shows the characteristics of the participants (5070 men and 6565 women) according to the sex-specific grip strength. Men in the LGS group had lower height (167.5 ± 0.9 cm vs. 172.5 ± 0.1 cm, *p* < 0.001), weight (66.6 ± 1.5 kg vs. 73.2 ± 0.2 kg, *p* < 0.001), and blood pressure (systolic, 114.5 ± 1.3 mmHg vs. 118.3 ± 0.3 mmHg, *p* = 0.004; diastolic, 76.2 ± 0.9 mmHg vs. 79.1 ± 0.2 mmHg, *p* = 0.003) than those in the NGS group. In women, the LGS group had lower height (157.3 ± 0.5 cm vs. 159.6 ± 0.1 cm, *p* < 0.001) and weight (56.3 ± 0.7 kg vs. 58.0 ± 0.1 kg, *p* = 0.026) than the NGS group. In both men and women, there was a significant difference in the distribution of household income between the LGS and NGS groups. The rate of alcohol consumption among men in the LGS group was no consumption (54.7%), low consumption (36.6%), and high consumption (8.7%), whereas the NGS group had low consumption (51.6%), high consumption (31.3%), and no consumption (17.2%), showing a significant difference in distribution between the two groups (*p* < 0.001). Almost half (46.2%) of the LGS group were nonsmokers, 29.1% were current smokers, and 24.8% were ex-smokers. In the NGS group, 40.6%, 31.8%, and 27.6% were current smokers, ex-smokers, and nonsmokers, respectively. There was a significant difference in the smoking status between the two groups (*p* < 0.001). The health-related behavior of women was not significantly different between the two groups.

In men, the mean HGS (23.8 ± 0.4 kg vs. 43.4 ± 0.1 kg, *p* < 0.001), HGSWR (37.2 ± 1.1 kg vs. 60.3 ± 0.2 kg, *p* < 0.001), absolute HGS (47.2 ± 1.1 kg vs. 83.9 ± 0.3 kg, *p* < 0.001), and relative HGS (2.1 ± 0.1 m^2^ vs. 3.5 ± 0.0 m^2^, *p* < 0.001) showed significant differences between the two groups. Likewise, in women, the mean HGS (15.8 ± 0.1 kg vs. 26.3 ± 0.1 kg, *p* < 0.001), HGSWR (28.8 ± 0.4 kg vs. 46.0 ± 0.1 kg, *p* < 0.001), absolute HGS (31.7 ± 0.3 kg vs. 50.4 ± 0.1 kg, *p* < 0.001), and relative HGS (1.4 ± 0.0 m^2^ vs. 2.3 ± 0.0 m^2^, *p* < 0.001) showed significant differences between the two groups.

### 3.2. Nutrients Intake of the Participants

Table 2 shows the analysis of the nutrient intake of the participants. In men, there was no significant difference in energy intake (2232 ± 120 kcal/day vs. 2391 ± 31 kcal/day) between the two groups, but the proportion of energy from carbohydrates (66.5 ± 1.5% vs. 63.5 ± 0.4%, *p* = 0.048) was significantly higher in the LGS group than in the NGS group, whereas the proportion of energy from protein (13.8 ± 0.5% vs. 15.4 ± 0.2%, *p* = 0.001) was significantly lower in the LGS group than in the NGS group. Intakes of protein (74.1 ± 5.1 g/day vs. 86.7 ± 1.5 g/day, *p* = 0.016), vitamin B2 (1.4 ± 0.1 mg/day vs. 1.7 ± 0.0 mg/day, *p* = 0.024), niacin (16.1 ± 1.3 mg/day vs. 19.2 ± 0.4 mg/day, *p* = 0.015), and iron (16.0 ± 1.2 mg/day vs. 19.0 ± 0.5 mg/day, *p* = 0.006) in the LGS group were significantly lower than that in the NGS group. In women, energy intake, energy from macronutrients, and micronutrient intake did not show a significant difference according to HGS.

Inadequate nutrient intake was analyzed by comparing the nutrient intake of participants with the EAR. The proportion of male participants with inadequate intakes of protein (55.1% vs. 36.1%, *p* < 0.001), vitamin A (64.3% vs. 53.2%, *p* = 0.046), vitamin B1 (19.4% vs. 8.4%, *p* < 0.001), vitamin B2 (52.8% vs. 36.5%, *p* = 0.002), niacin (45.6% vs. 24.1%, *p* < 0.001), phosphorus (17.6% vs. 6.4%, *p* < 0.001), and iron (19.4% vs. 9.6%, *p* = 0.001) in the LGS group was significantly higher than that in the NGS group. In women, only the proportion of participants with inadequate protein intake in the LGS group (57.4% vs. 47.2%, *p* = 0.004) was significantly higher than that in the NGS group. There was no significant difference between the other nutrients and HGS.

Nutrient density was analyzed by calculating the number of nutrients per 1000 kcal (data not shown). In men, the densities of protein (30.8 ± 1.2 g/1000 kcal/day vs. 36.1 ± 0.4 g/1000 kcal/day, *p* < 0.001), niacin (6.6 ± 0.5 mg/1000 kcal/day vs. 7.9 ± 0.1 mg/1000 kcal/day, *p* = 0.004), phosphorus (472.2 ± 19.2 mg/1000 kcal/day vs. 526.0 ± 5.7 mg/1000 kcal/day, *p* = 0.005), and iron (6.9 ± 0.5 mg/1000 kcal/day vs. 8.0 ± 0.2 mg/1000 kcal/day, *p* = 0.009) in the LGS group were significantly lower than that in the NGS group. There was no significant difference in nutrient density according to HGS in women.

### 3.3. Identification of Dietary Patterns

Factor analysis of the 25 food items was performed using factor loading with three main dietary patterns for each sex (Appendix A). Dietary patterns were derived from different results based on eigenvalue and food items. In men, three patterns were extracted: “Westernized Korean dietary pattern (Factor 1)”, “Convenience dietary pattern (Factor 2)”, and “Traditional Korean dietary pattern (Factor 3)”. A Westernized Korean dietary pattern was characterized by a high intake of seasonings, oils, meats, sugar and sweets, fish and shellfish, and non-sugar beverages. A convenience dietary pattern was characterized by a high intake of flour and bread, milk and dairy products, soda, processed food, and eggs, whereas a Traditional Korean dietary pattern was characterized by a high intake of white rice, whole grains, legumes, nuts, fruits and vegetables, kimchi, and seaweeds. In women, three dietary patterns were also identified, “Westernized Korean dietary pattern (Factor 1)”, “Traditional Korean dietary pattern (Factor 2)”, and “Healthy dietary pattern (Factor 3)”. A Westernized Korean dietary pattern was characterized by a high intake of oils, seasonings, sugar and sweets, non-sugar beverages, fish and shellfish. A Traditional Korean dietary pattern was characterized by a high intake of white rice, legumes, kimchi, and seaweeds, whereas a healthy dietary pattern was characterized by a high intake of nuts, fruits and vegetables, and whole grains.

### 3.4. ORs and 95% CIs for LGS According to Inadequacy of Dietary Intake

The ORs and 95% CIs of the inadequacy of dietary intake for LGS are shown in Table 3. After adjusting for age and energy intake (Model 1), inadequate protein intake was associated with an increased risk of LGS in both men (OR 2.114, 95% CI 1.358–3.289) and women (OR 1.589, 95% CI 1.207–2.092). In the same model, some micronutrients were associated with an increased risk of LGS in men. However, after adjusting for other covariates such as physical activity, household income, smoking status, alcohol consumption, and energy intake (Model 2), it was found that inadequate intakes of vitamin B1 (OR 2.916, 95% CI 1.265–6.719), niacin (OR 2.286, 95% CI 1.095–4.774), phosphorus (OR 2.731, 95% CI 1.036–7.199), and iron (OR 2.591, 95% CI 1.102–6.088) were associated with an increased risk of LGS in men. In women, only inadequate protein intake was associated with an increased risk of LGS (OR 1.976, 95% CI 1.248–3.127, *p* = 0.004).

### 3.5. ORs and 95% CIs for LGS According to Dietary Patterns

Table 4 shows the association between dietary patterns and LGS. In the crude and Model 1, a more Westernized Korean dietary pattern was significantly associated with a decreased risk of LGS (crude OR 0.323, 95% CI 0.175–0.596; Model 1 OR 0.386, 95% CI 0.190–0.782) in men compared with a less Westernized Korean dietary pattern, but there was no significance in Model 2, which adjusted other covariates such as BMI, physical activity, household income, smoking status, and alcohol intake. None of the three dietary patterns in women was significantly associated with LGS.

## 4. Discussion

This study investigated the relationship between dietary factors and LGS by sex in a young and middle-aged population, based on nationally representative data from South Korea. In men, the LGS group had a higher proportion of energy from carbohydrates and a lower proportion of energy from protein than the NGS group. The LGS group consumed less protein, vitamin B2, niacin, and iron than the NGS group, and their diet had lower protein, niacin, phosphorus, and iron densities than that of the NGS group. Insufficient intake of vitamin B1, niacin, phosphorus, and iron increased the risk of LGS. In women, only insufficient protein intake increased the risk of LGS.

As protein is an essential nutrient for muscle health, many studies have reported that adequate protein intake helps prevent and manage sarcopenia [30,31,32]. Among men in the present study, the protein intake was lower in the LGS group than in the NGS group as we expected. In addition, there was a significantly lower proportion of energy from protein in the LGS group. Although the average protein intake in the LGS group (1.1 g/kg) was higher than the EAR of protein for Korean adults (0.73 g/kg/day), the proportion of the male participants with insufficient protein intake less than the EAR in the LGS group was 1.5 times higher than in the NGS group. In addition, there were some opinions being raised that the EAR of protein should be elevated to prevent sarcopenia and muscle weakness; thus, we needed to consider it in interpreting the protein intake level [29]. Conversely, the proportion of energy from carbohydrates in the LGS group (67 %) was higher than that in the NGS group. It exceeded the 55%–65% recommended level by KDRIs [29]. This result is comparable to the studies that high HGS is associated with low prevalence of metabolic syndrome because high carbohydrate intake can increase the risk of metabolic syndrome such as diabetes and obesity [33,34]. Because dietary carbohydrate stimulates insulin secretion, high carbohydrate intake can induce body fat accumulation, which can affect body composition [35]. Especially, as carbohydrate portion is higher in the Korean diet than in the Western diet, further research is needed regarding this issue.

Vitamins and minerals are effective for muscle, bone, and neuromuscular health, and studies have found that a high intake of B vitamins and minerals is associated with high HGS [36,37]. Especially, B vitamins are involved in energy and protein metabolism [38]. Among the B vitamins, vitamin B1 is a coenzyme of pyruvate dehydrogenase related to carbohydrate metabolism and neural integrity and function [38]. In addition, niacin is known to be involved in energy metabolism and is associated with coenzymes [39]. B vitamins and minerals such as phosphorus and iron have been reported to affect energy metabolism and muscle health, but the results are not consistent [40,41,42,43,44,45,46,47,48,49,50]. Among men in the present study, the average intake of these nutrients was higher than the EAR in the LGS group. However, the proportion of men with insufficient intake less than EAR in the LGS group was twice as much as the NGS group, which means that the personal variation of these nutrients intake was high in the LGS group. Thus, long-term insufficient intake of these nutrients is thought to affect energy metabolism and exercise capacity.

Among women in the present study, insufficient protein intake increased the risk of LGS. This result is consistent with a previous cross-sectional study, which found that higher protein intake was associated with higher muscle mass [51], and a prospective study found that protein intake was associated with the preservation of grip strength in adults aged 29–85 years [52]. Another study observed longitudinal changes in skeletal muscle mass (SMM) and reported that the higher the protein intake, the higher the SMM, which was stronger in women than in men. The study also found that women with a low animal protein intake had a lower total protein intake than the recommended daily allowance [53]. In addition, according to the National Food and Nutrition Statistics analyzing the nutrient intake of Koreans, in women, 40–60% of the protein intake was from plant protein [54]. Although the type of protein was not analyzed in the present study, considering the previous studies and extracted dietary patterns, it is thought that the women in the present study also had a plant protein-based diet and thus had insufficient animal protein intake. Therefore, insufficient protein intake and plant protein-based diet may increase the risk of LGS.

Because it is difficult to determine the correlation between a single nutrient and muscle strength, several studies have investigated the association between dietary patterns and skeletal muscle health [18,19]. In these studies, prudent and Mediterranean diets were positively related to skeletal muscle health. A prudent diet emphasizes the intake of fruits, vegetables, whole grains, low-fat and fat-free dairy, healthy fats, lean meats, and poultry while limiting calories from added sugars and refined starches [55]. The Mediterranean diet emphasizes the intake of fruits, vegetables, potatoes, whole grains, beans, nuts, seeds, and extra virgin olive oil [56]. These healthy dietary patterns are believed to have a myoprotective effect because a healthy diet has an antioxidant effect that eliminates reactive oxygen species involved in muscle atrophy as well as positive effects on reducing the risk of chronic disease [57]. However, Western and Asian populations exhibit different dietary patterns. In particular, the Korean diet is characterized by higher carbohydrates and lower fat than the Western diet. Especially, because the fat proportion in the Korean diet is less than 20%, it is hard to compare the Korean dietary pattern with the Mediterranean. Therefore, we extracted new dietary patterns from representative data and analyzed the association with LGS. These new patterns are thought to reflect the Korean dietary pattern better. However, these new patterns in the present study had no significant correlation with LGS in men and women. In men, the Westernized Korean dietary pattern correlated with decreased LGS risk after adjusting for age and energy intake, but there was no significance when other covariates were additionally adjusted. In women, there was no significant correlation between dietary patterns and HGS. The Westernized Korean dietary pattern includes not only myoprotective foods such as fish and shellfish and oil but also non-myoprotective foods such as sugar and sweets and seasoning. In addition, this dietary pattern does not include representative myoprotective foods such as fruits and vegetables, whole grains, legumes and nuts. On the contrary, the Traditional Korean dietary pattern includes myoprotective foods such as whole grains, legumes, nuts, fruit and vegetables but does not include meats, fish and shellfish, and eggs, which are the main sources of animal protein. The Korean diet is dominated by plant protein sources that lack one or more essential amino acids, are of low biological value, and can lead to protein deficiency and poor muscle health. To overcome this, the protective effect of muscle health should be increased by increasing the intake of animal sources of protein such as lean meats, fish and shellfish that have high biological value and contain all essential amino acids [52].

In this study, the association between dietary factors and LGS differed by sex. More nutrients were involved in LGS of men than of women. This may imply that the quality of diet in men is poorer than women because the diet composition of men has less various foods than women. Actually, in a survey, the healthy eating index of Korean adults was lower in men (60.9) than in women (63.8) [58]. In addition, the difference of changes in body composition by sex may affect the result. In one study, men’s muscle mass peaked in their 30s and then gradually decreased with age, whereas in women, muscle mass increased until their 40s, was maintained in their 50s, and then decreased thereafter [59].

This study has several limitations. First, it is a cross-sectional study; hence, it is difficult to identify a causative relationship between dietary factors and HGS clearly. Second, dietary intake was calculated on the basis of a 24 h recall method, which might not have been representative of participants’ typical intake. In addition, since it is a self-reported method, nutrients that are affected by external factors such as vitamin D from sun exposure were not considered. However, at the population level, it can provide rich details about mean dietary intake for a given day. Despite these limitations, our study has several important strengths. First, it was based on a nationally representative sample in south Korea. Second, this is the first study to investigate the sex-specific association between nutrients, nutritional density, dietary pattern, and low muscle strength in young and middle-aged Koreans.

## 5. Conclusions

LGS is correlated with insufficient intake of vitamin B1, niacin, phosphorus, and iron in men and insufficient protein intake in women. In men, the Westernized Korean dietary pattern correlated with decreased LGS risk after adjusting for age and energy intake, but there was no significance when other covariates were additionally adjusted. In women, there was no significant correlation between dietary patterns and HGS. Sex-specific dietary intake may affect HGS in young and middle-aged adults in Korea. Further research to confirm the relationship between these factors and LGS and the mechanism of this relationship is pertinent.

## Figures and Tables

**Table 1 nutrients-14-03816-t001:** Sociodemographic characteristics of the participants.

Variables	Men (*n* = 5070, wn = 59,336,338)			Women (*n* = 6565, wn = 51,774,435)		
	LGS (*n* = 111, wn = 1,175,179)	NGS (*n* = 4959, wn = 58,1161,159)	*p* Value	LGS (*n* = 305, wn = 2,354,354)	NGS (*n* = 6260, wn = 49,420,3081	*p* Value
Age (year) ^1^	40.3 ± 1.5	40.2 ± 0.2	0.952	40.5 ± 1.1	39.9 ± 0.2	0.625
Height (cm) ^1^	167.5 ± 0.9	172.5 ± 0.1	<0.001	157.3 ± 0.5	159.6 ± 0.1	<0.001
Body weight (kg) ^1^	66.6 ± 1.5	73.2 ± 0.2	<0.001	56.3 ± 0.7	58.0 ± 0.1	0.026
BMI (kg/m^2^) ^1^	23.7 ± 0.5	24.6 ± 0.1	0.097	22.7 ± 0.3	22.8 ± 0.1	0.967
WC (cm) ^1^	82.8 ± 1.4	85.4 ± 0.2	0.052	76.2 ± 0.7	76.1 ± 0.2	0.851
Systolic BP (mmHg) ^1^	114.5 ± 1.3	118.3 ± 0.3	0.004	109.9 ± 1.0	110.2 ± 0.2	0.792
Diastolic BP (mmHg) ^1^	76.2 ± 0.9	79.1 ± 0.2	0.003	71.6 ± 0.6	72.7 ± 0.2	0.072
HGS (kg) ^1^	23.8 ± 0.4	43.4 ± 0.1	<0.001	15.8 ± 0.1	26.3 ± 0.1	<0.001
HGSWR ^1^	37.2 ± 1.1	60.3 ± 0.2	<0.001	28.8 ± 0.4	46.0 ± 0.1	<0.001
Absolute HGS (kg)	47.2 ± 1.1	83.9 ± 0.3	<0.001	31.7 ± 0.3	50.4 ± 0.1	<0.001
Relative HGS (m^2^)	2.1 ± 0.1	3.5 ± 0.0	<0.001	1.4 ± 0.0	2.3 ± 0.0	<0.001
Household income ^2^						
Q1	35 (30.3)	378 (7.7)	<0.001	50 (14.8)	484 (7.6)	<0.001
Q2	29 (22.5)	1102 (22.1)		85 (25.5)	1489 (23.6)	
Q3	28 (25.5)	1606 (32.6)		81 (29.1)	2036 (32.4)	
Q4	18 (21.7)	1860 (37.6)		89 (30.6)	2231 (36.5)	
Residential area ^2^						
Urban	90 (86.2)	4139 (86.2)	0.997	247 (85.1)	5372 (88.2)	0.179
Rural	21 (13.8)	820 (13.8)		58 (14.9)	878 (11.8)	
Alcohol consumption ^2^						
None	46 (54.7)	601 (17.2)	<0.001	99 (35.5)	1570 (29.9)	0.126
Low	30 (36.6)	1688 (51.6)		147 (59.1)	3014 (61.7)	
High	6 (8.7)	1074 (31.3)		13 (5.4)	386 (8.4)	
Smoking status ^2^						
Nonsmoker	42 (46.2)	1244 (27.6)	<0.001	270 (89.0)	5396 (87.5)	0.519
Ex-smoker	29 (24.8)	1649 (31.8)		16 (7.0)	401 (6.8)	
Current smoker	35 (29.1)	1929 (40.6)		14 (4.0)	320 (5.7)	
Physical activity ^2^						
Low (<600 MET-min/week)	89 (77.1)	3417 (67.4)	0.058	252 (80.3)	4855 (76.8)	0.320
Moderate (600–3000 MET-min/week)	17 (20.5)	1198 (25.3)		47 (17.5)	1189 (19.6)	
High (>3000 MET-min/week)	5 (2.4)	344 (7.2)		6 (2.1)	216 (3.7)	

LGS, low HGS; NGS, normal handgrip strength; BMI, body mass index; WC, waist circumference; BP, blood pressure; HGS, handgrip strength; HGSWR, handgrip-strength-to-weight ratio; Q, quartile; wn, weighted number. Bold indicates significant at *p* value < 0.05. ^1^ Values are least square mean ± standard error. ^2^ Values are number (weighted %).

**Table 2 nutrients-14-03816-t002:** Assessment of dietary intake and adequacy of nutrients intake of the participants.

Variables	Men (*n* = 5070, wn = 59,336,338)			Women (*n* = 6565, wn = 51,774,435)		
	LGS (*n* = 111, wn = 1,175,179)	NGS (*n* = 4959, wn = 58,161,159)	*p* Value	LGS (*n* = 305, wn = 2,354,354)	NGS (*n* = 6260, wn = 49,420,081)	*p* Value
Energy intake and energy from macronutrients ^1^						
Energy (kcal/day)	2232 ± 120	2391 ± 31	0.188	1771 ± 53	1774 ± 28	0.944
Carbohydrate (% of energy)	66.5 ± 1.5	63.5 ± 0.4	0.048	62.6 ± 0.9	62.7 ± 0.5	0.868
Protein (% of energy)	13.8 ± 0.5	15.4 ± 0.2	0.001	15.7 ± 0.4	15.3 ± 0.2	0.245
Fat (% of energy)	19.7 ± 1.4	21.1 ± 0.3	0.287	21.7 ± 0.7	22.0 ± 0.4	0.597
Nutrients intake ^1^						
Protein (g/day)	74.1 ± 5.1	86.7 ± 1.5	0.016	66.1 ± 2.8	64.3 ± 1.4	0.448
Vitamin A (μgRE/day)	741.4 ± 121.7	774.0 ± 52.0	0.780	589.6 ± 63.3	610.0 ± 32.3	0.727
Vitamin B1 (mg/day)	2.0 ± 0.2	2.2 ± 0.0	0.260	1.6 ± 0.1	1.7 ± 0.0	0.246
Vitamin B2 (mg/day)	1.4 ± 0.1	1.7 ± 0.0	0.024	1.3 ± 0.1	1.3 ± 0.0	0.857
Vitamin C (mg/day)	88.1 ± 10.3	95.3 ± 3.2	0.495	84.4 ± 8.4	96.8 ± 5.0	0.088
Niacin (mg/day)	16.1 ± 1.3	19.2 ± 0.4	0.015	14.6 ± 0.6	14.7 ± 0.4	0.842
Calcium (mg/day)	513.0 ± 48.9	562.8 ± 11.1	0.309	439.1 ± 19.4	455.1 ± 11.0	0.340
Phosphorus (mg/day)	1121.8 ± 75.1	1271.8 ± 19.9	0.051	977.4 ± 33.1	982.6 ± 18.7	0.861
Iron (mg/day)	16.0 ± 1.2	19.0 ± 0.5	0.006	14.4 ± 0.8	14.7 ± 0.4	0.675
Inadequate nutrients intake (Below the EAR) ^2^						
Protein ^3^	65 (55.1)	1855 (36.1)	<0.001	169 (57.4)	2750 (47.2)	0.004
Vitamin A	76 (64.3)	2655 (53.2)	0.046	172 (57.3)	3298 (53.2)	0.232
Vitamin B1	20 (19.4)	409 (8.4)	<0.001	53 (16.5)	948 (16.0)	0.832
Vitamin B2	61 (52.8)	1843 (36.5)	0.002	121 (36.8)	2221 (35.2)	0.619
Vitamin C	75 (65.5)	3074 (63.2)	0.667	199 (65.6)	3655 (59.8)	0.086
Niacin	53 (45.6)	1210 (24.1)	<0.001	115 (36.2)	2269 (36.3)	0.976
Calcium	84 (74.75)	3341 (68.0)	0.184	224 (75.0)	4343 (69.4)	0.061
Phosphorus	20 (17.6)	305 (6.4)	<0.001	52 (16.3)	922 (15.1)	0.632
Iron	19 (19.4)	407 (9.0)	0.001	93 (34.7)	1865 (32.5)	0.537

LGS, low handgrip strength; NGS, normal handgrip strength; EAR, Estimated average requirement; RE, retinol equivalents; wn, weighted number. ^1^ Values are least square mean ± standard error adjusted for age, body mass index, physical activity, household income, smoking status, and alcohol consumption. ^2^ Values are presented as number (weighted %). ^3^ Defined as who consumed <0.73 g/kg/day, which is estimated average requirement in the Dietary Reference Intakes Koreans 2015.

**Table 3 nutrients-14-03816-t003:** Odds ratios for LGS according to inadequate nutrients intake.

Variables	Men						Women					
	Model 1		*p* Value	Model 2		*p* Value	Model 1		*p* Value	Model 2		*p* Value
	OR	95% CI		OR	95% CI		OR	95% CI		OR	95% CI	
Protein	2.114	1.358–3.289	<0.001	1.525	0.646–3.597	0.335	1.589	1.207–2.092	0.001	1.976	1.248–3.127	0.004
Vitamin A	1.269	0.774–2.082	0.345	1.554	0.803–3.006	0.191	1.185	0.885–1.588	0.254	0.953	0.653–1.390	0.802
Vitamin B1	2.001	1.076–3.724	0.029	2.916	1.265–6.719	0.012	1.012	0.669–1.532	0.953	0.898	0.518–1.559	0.703
Vitamin B2	1.547	0.907–2.640	0.109	1.817	0.958–3.445	0.067	1.042	0.741–1.464	0.814	0.897	0.597–1.347	0.600
Vitamin C	0.944	0.580–1.538	0.817	0.603	0.323–1.128	0.113	1.313	0.992–1.738	0.057	1.252	0.874–1.795	0.221
Niacin	2.362	1.386–4.024	0.002	2.286	1.095–4.774	0.028	0.946	0.681–1.315	0.742	0.939	0.627–1.408	0.762
Calcium	1.052	0.634–1.746	0.844	1.754	0.863–3.563	0.120	1.349	0.962–1.890	0.082	1.189	0.804–1.758	0.386
Phosphorus	2.390	1.171–4.878	0.017	2.731	1.036–7.199	0.042	1.059	0.694–1.616	0.789	1.143	0.666–1.959	0.628
Iron	1.902	1.003–3.604	0.049	2.591	1.102–6.088	0.029	1.152	0.814–1.632	0.424	0.956	0.626–1.458	0.833

OR, odds ratio; CI, confidence interval. Reference values are the normal handgrip strength group. Model 1 was adjusted for age and energy intake. Model 2 was adjusted for age, body mass index, physical activity, household income, smoking status, alcohol intake, and energy intake.

**Table 4 nutrients-14-03816-t004:** Odds ratios for LGS according to tertiles of dietary patterns.

Variables	Crude			*p* for Trend	Model 1			*p* for Trend	Model 2			*p* for Trend
	OR (95% CI)				OR (95% CI)				OR (95% CI)			
	T1	T2	T3		T1	T2	T3		T1	T2	T3	
Men												
Westernized Korean	1.000	0.785 (0.175–0.596)	0.323 (0.175–0.596)	0.002	1.000	0.843 (0.515–1.380)	0.386 (0.190–0.782)	0.030	1.000	0.887 (0.486–1.619)	0.506 (0.217–1.175)	0.285
Convenience	1.000	0.848 (0.503–1.403)	0.772 (0.430–1.210)	0.458	1.000	0.786 (0.462–1.338)	0.800 (0.468–1.368)	0.588	1.000	0.747 (0.372–1.501)	0.684 (0.331–1.414)	0.550
Traditional Korean	1.000	0.728 (0.421–1.257)	1.092 (0.667–1.787)	0.347	1.000	0.719 (0.405–1.276)	1.311 (0.724–2.373)	0.121	1.000	0.524 (0.258–1.064)	0.830 (0.408–1.689)	0.185
Women												
Westernized Korean	1.000	0.880 (0.609–1.273)	0.929 (0.678–1.274)	0.784	1.000	0.902 (0.612–1.330)	0.989 (0.643–1.521)	0.847	1.000	0.919 (0.609–1.387)	1.032 (0.674–1.645)	0.860
Traditional Korean	1.000	1.119 (0.805–1.555)	1.046 (0.732–1.495)	0.791	1.000	1.096 (0.784–1.532)	1.027 (0.722–1.463)	0.851	1.000	1.015 (0.706–1.461)	0.932 (0.625–1.389)	0.887
Healthy	1.000	0.896 (0.660–1.217)	0.946 (0.705–1.319)	0.778	1.000	0.858 (0.625–1.177)	0.929 (0.626–1.275)	0.636	1.000	0.809 (0.565–1.160)	0.827 (0.571–1.200)	0.462

OR, odds ratio; CI, confidence interval. Model 1 was adjusted for age and energy intake. Model 2 was adjusted for age, body mass index, physical activity, household income, smoking status, alcohol consumption, and energy intake.

## Data Availability

Data are available at http://knhanes.cdc.go.kr/.

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
