# Peer review of "Sex and Diet-Related Disparities in Low Handgrip Strength among Young and Middle-Aged Koreans: Findings Based on the Korea National Health and Nutrition Examination Survey (KNHANES) from 2014 to 2017"

_nutrients, 2022, doi:10.3390/nu14183816_

Round 1

Reviewer 1 Report

dear authors,

the title can be improved further to be informative

keywords are repetitive from title use mesh library to explore further https://meshb.nlm.nih.gov/

abstract is adequate

the introduction provided state of the art description of the issue under study, however there is no clear objective of the present study. authors also did not provide any hypothesis implying that this study was purely exploratory.

in the introduction I urge the the authors to reorder neutrients when the present them as macro, micro.

the study population correctly excluded people e.g. cancer, dm etc but authors need to provide justification by referring to studies that show variation between health and general population hsg. also why authors did not exclude people who might have higher hsg e.g. people with hepa activity.

authors are encouraged to explain more about the knhhanes study as readers outside Korea not familiar with it. 

section 2.2 need to be rewitten for details e.g. lines 89-91 clearly talk about ipaq but authors did not provide explicit info.

section 2.3 is well argued. 

section 2.4 did you address supplements intake? did u consider sun exposure for vit D? If not kindly add to limitations

section 2.5 lines 122-129 ok lines 130-135 move to methods

line 130 which extraction method was used only rotation technique was presented. also add KMO and Bartlett tests results. provide references for these too.

section 2.6 I am aware of large sample size but did u consider 1)reporting sample size/power analysis and 2) did u perform normality testsb (see my comment below on Table 1)

some results reported has no methods e.g. what is low vs high alcohol consumption?

table 1 show that there is problem with normality and results does not make sense e.g. height and weight both significant for lgs and ngs (male and female) but bmi is not. I would like to see 95%ci for all reaults pls

hgs and hgswr reported in table 1 has no meaning table 1 was suppose to be lgs and ngs

table 1 at end better use same language as methods and refer to met instead of inactive, active, hepa

table 2 is so busy. there is clear power issue as lgs far less than ngs in both men and women

table 3 has no meaning statistically or theoritically pls replace it with simple anova test. table 5 perform comparisons also so maybe u can drop entire table 3

table 4&5 adequate . see my comment regarding table 3. 

study limitations need to be revised in light of suggestions above. the study is bases on self report.

conclusion need to be toned down little. 

Reviewer 2 Report

The strength of study is the use of micronutrients in relation to GS in a large sample. However, two concerns are the use of sarcopenia cut off with young adults and the use of dietary patterns that is not well supported by literatures. This causes a very weak introduction and discussion, and result can also be improved and regrouped. Therefore, the paper requires further reanalysis and rewriting many sections.

1-      The study does not include old population. Thus, the introduction and aim of study must be rewritten. The same objective from the same data source in old population has already been done in reference 6. Doing the same study on younger (lower than 65 years) does not add any value, and using the term sarcopenia in young is questionable and arguable.

2-      Introduction: Paragraph 1 and 2 must be rewritten to highlight micronutrients and health and strength, and to highlight dietary patterns, and to reduce the topic of sarcopenia and nutrition in old population. The rational of study also needs to be reconsidered. There are some comments in the introduction as follows:

a.       Line 44, ref 11 did not say that. The paper is on the relationship between grip strength and MetS and quality of life.

b.       Line 53, provide details of reference 6 which is the same source of your data, and reference 13 is a general brief review.

c.       Line 56, you did not use reference 10 properly. This reference found increased CHO intake and decreased protein intake in older population

3-      Analysis and result:

a.       As the sample is adults, LGS group in men was 111 participants only (2%) whereas the control group (normal grip strength) is five thousand. The use of sarcopenia cut off is not appropriate in the current healthy young samples. I would prefer to use tertiles or quartiles to better utilize the large sample wisely. Grip strength is used as a sarcopenia index, but its relation to metabolic syndrome and health is wider than sarcopenia. Thus, you can discuss hand grip strength out of the context of sarcopenia. By the way, grip strength among males and females has been studies for a decade ago (e.g. Urska Puh, 2009),

b.       I like the use of dietary patterns, and this is one of the strengths of study (e.g. Table 5). However, you used patterns that are not supported by the literatures. Therefore, you struggled in the discussion, and comes up with contradictive and unproved conclusion in line 377. My advice is that you keep these patterns and include Prudent and Mediterranean diet patterns to the result. Why do not you include old population in the analysis of dietary patterns? This will really strengthen the study very much.

4-      Discussion: it is too long and too wide. Paragraph 4 should be reduced, and part of it is out of context, and is not related to result. Paragraph 6 will be appropriate if you add these dietary patterns to the result, otherwise how you discuss topic that is not appeared in your result. Paragraph 7 is not related to the result, and does not add any benefit to the discussion.

Round 2

Reviewer 1 Report

thank you for addressing my concerns.

Author Response

Many thanks. We appreciate your time.

We checked the spelling as a whole.

Reviewer 2 Report

Line 64 : What the limitation is and what you will add. Justify why your dietary approach is worthdoing. The sentence in line 66 is not enough. 

Your answer on point 7 in the 1st round can build an argument, but I could not see it in your introduction and discussion. Namely, you can use your response on the comment and these references wisely to see it in the manuscript. 

Line 321 : you cant say "insufficient protein intake in LGS". Maybe just lower than NGS. 

Lines 337-349 : how does this link to your result? Either connect it with your result or hugely minimize it. 

In general, try to connect the whole paragraph with your result as much as possible, and delete sentences that are not related. 

Line 356 : Prove it or delete it. A huge claim!

364 - 366 : I am trying to connect these sentences with the dietary patterns that you used in your study. You just put sentences that are slightly related to the topic. There is not argument. 

Line 392 - 394 : Your conclusion "Therefore, it is thought that a dietary pattern in which “Westernized Korean pattern” and “Traditional Korean pattern” are properly harmonized is recommended for Korea adults for preventing muscle weakness." is different from your result in line 294 - 299 !!

I understand your scientific concern about Westernized diet, but you have to put an assumption that is not examined as a conclusion.

Line 416 : do not use associated. Maybe correlated! 

You did not mention dietary patterns in the conclusion. This is very important point.  
